# The Janus Face of Tumor Microenvironment Targeted by Immunotherapy

**DOI:** 10.3390/ijms20174320

**Published:** 2019-09-03

**Authors:** Maria Buoncervello, Lucia Gabriele, Elena Toschi

**Affiliations:** 1Research Coordination and Support Service, Istituto Superiore di Sanità, 00161 Rome, Italy; 2Tumor Immunology Section, Department of Oncology and Molecular Medicine, Istituto Superiore di Sanità, 00161 Rome, Italy

**Keywords:** tumor microenvironment (TME), tumor invasion, pre-metastatic niche (PMN), immune cells, immunotherapy

## Abstract

The tumor microenvironment (TME) is a complex entity where host immune and non-immune cells establish a dynamic crosstalk with cancer cells. Through cell-cell interactions, which are mediated by key signals, such as the PD-1/PD-L1 axis, as well as the release of soluble mediators, this articulated process defines the nature of TME determining tumor development, prognosis, and response to therapy. Specifically, tumors are characterized by cellular plasticity that allows for the microenvironment to polarize towards inflammation or immunosuppression. Thus, the dynamic crosstalk among cancer, stromal, and immune components crucially favors the dominance of one of the Janus-faced contexture of TME crucial to the outcome of tumor development and therapeutic response. However, mostly, TME is dominated by an immunosuppressive landscape that blocks antitumor immunity and sustain tumor progression. Hence, in most cases, the immunosuppressive components of TME are highly competent in suppressing tumor-specific CD8^+^ T lymphocytes, the effectors of cancer destruction. In this complex context, immunotherapy aims to arm the hidden Janus face of TME disclosing and potentiating antitumor immune signals. Herein, we discuss recent knowledge on the immunosuppressive crosstalk within TME, and share perspectives on how immunotherapeutic approaches may exploit tumor immune signals to generate antitumor immunity.

## 1. Introduction

Recent advances in the tumor microenvironment (TME) composition have uncovered the extensive heterogeneity of this site for multiple cellular components, variable states of their differentiation and plastic cell functions. Hence, TME includes a broad range of cells that diverge in ontogeny, phenotypic and functional characteristics, immune interactions, tumor propagation potential, and response to therapies [1]. This complex entity comprises neoplastic cells at different stage of differentiation, including cancer stem cells (CSCs) and epithelial and stromal cells, such as cancer-associated fibroblasts (CAFs), various infiltrating immune cells, and non-cell components of extracellular matrix (ECM). A complex array of reciprocal signaling among all of these components defines a dynamic immunosuppressive tumor niche, which fuels tumor growth and invasion and therapy resistance [2]. Therefore, TME composition is strictly associated with the clinical outcome of cancer patients to the pint that the analysis of tumor components has become fundamental to predict the response to treatment. Over the last few years, the growing knowledge of the dynamic signals within TME has led to the concept that this niche may be reeducated to generate antitumor immunity changing the fate of cancer cells. Thus, a big challenge is to develop new therapeutic strategies that are able to control the dynamic crosstalk among the cells within TME towards an efficient blocking of immunosuppressive signals. In this light, this review provides an overview of the major components that drive tumor progression and examines the dynamic crosstalk among tumor, stromal cells, and their products playing a crucial role in determining the recruitment, composition, and function of immune-infiltrating cells [3]. Lastly, the major immunotherapeutic strategies that are designed to target active TME signals for reversing immunosuppression into antitumor immunity will be discussed.

## 2. The Dynamic Niche of TME

During tumor development, a remodeling of the tissue occurs, which implies the modification of ECM and the involvement of stromal cells, such as CAFs, endothelial cells (ECs), pericytes, adipocytes, activated tissue fibroblasts, mesenchymal stem cells (MSCs), and tumor-infiltrating immune cells [4,5]. This heterogenous microenvironment is known as TME (Figure 1). 

### 2.1. The Role of ECM

The complex interactions between tumor cellular components and ECM may directly or indirectly influence the main hallmarks of cancer cells, through the induction of apoptosis, migration, and proliferation, also depending on the type of tumor and its localization. 

The ECM is an intricate network that is composed by a variety of components such as collagen, integrins, laminin, fibronectin, glycosaminoglycans, matrix metalloproteinases (MMPs), and secreted acidic proteins that are rich in cysteine that offer structural support, as well as biochemical and biomechanical signals, for cancer cell growth [6]. It has been hypothesized that the ECM might have both pro-angiogenic and vascular-stabilizing roles. In fact, it contains several pro-angiogenic growth factors, such as vascular endothelial growth factor A (VEGFA), fibroblast growth factors (FGFs), platelet-derived growth factor-beta (PDGF-β), and transforming growth factor-beta (TGF-β), which are proteolytically activated by plasmin, MMPs, or other proteases that are released by stromal and/or cancer cells [7,8]. In addition, the altered geometry and increased density of the collagen fibres may affect tumor angiogenesis [9,10]. It has been observed that ECM rigidity modulates the spatial organization of VEGFA gradients and VEGF receptor 2 expression by ECs. Moreover, the irregular organization of ECM fibres facilitates tumor angiogenesis by enhancing the migration of ECs, CAFs, and tumor-associated macrophages (TAMs). As expected, these cells more rapidly migrate on linearized collagen fibres, which are highly present in tumors as compared with non-neoplastic tissues [11]. Additionally, the integrins, which are the main receptors of ECM adhesion molecules playing a key role in regulating cell motility in physiological processes, such as development and wound healing [12], are strongly involved in cancer invasion and dissemination. These molecules can be implicated in all stages of cancer progression, from primary tumor formation to cancer cell extravasation and formation of the metastatic niche due to their adaptable functional properties [13]. 

### 2.2. The Contribution of Stromal Cells

In complex landscape of TME, tumor growth is mainly promoted through the secretion of multiple growth factors, MMPs, and factors that are produced by CAFs and/or tumor cells whose complex interplay stimulates stemness and epithelial-mesenchymal transition (EMT). Within TME, CAFs can differentiate from different type of cells, including fibroblasts, epithelial cells, ECs, and MSCs that under the influence of TGF-β, insulin-like growth factor (IGF) and epidermal growth factor (EGF), acquire functions as hyperactivation, increased proliferation, motility, and ECM production [6,14]. In turn, by secreting collagen type I and III, fibronectin, proteoglycans, and glycosaminoglycans, CAFs promote cancer cell migration and inhibit vascularization, thus contributing to EMT, cancer invasion, angiogenesis, and metastasis [6]. Furthermore, CAFs promote tumor progression through the release of interleukin-1 (IL-1), IL-6, IL-22, and IL-8, of which this latter plays a crucial role in hypoxia-induced tumor apoptosis resistance and EMT remodeling [15,16,17]. Under certain stimuli, CAFs acquire a pro-inflammatory signature and express immunomodulatory molecules (e.g., TGF-β or programmed cell death-ligand (PD-L1/L2), as well as chemokines (e.g., CXCL12, CCL2, CCL3, CCL4, and CCL5), which, in turn, stimulate the recruitment of immunosuppressive myeloid cells [3]. Among the others, CCL2, CCL3, and CCL5 contribute to tumor vascularization and metastasis and also stimulate MMP9 secretion by monocytes, which strongly favors tumor cell extravasation [18]. CCL2 and CCL5 also promote cancer cell growth, survival, migration and EMT. Of note, the key role of CCL2 in cancer progression has been demonstrated by the direct correlation between its overexpression and poor prognosis characterizing many types of primary tumor. Of interest, it has also been found that CAFs, in a dynamic interaction with tumor cells and stromal matrix, participate in the creation of a physical barrier for cytotoxic immune cell infiltration, known as a desmoplastic reaction, which surrounds the tumor nests [19]. Furthermore, although CAFs are the major source for VEGFA in tumors, they can support tumor angiogenesis in a VEGFA-independent manner [20]. To this regard, it has been observed that in melanoma, matured CAFs secrete the WNT antagonist frizzled-related protein 2, which impairs the malignant behavior of the tumors in old patients [21]. Overall, CAFs accumulation in TME is often associated with an immunosuppressive niche supporting tumor progression and poor prognosis [17]. Along with CAFs, another relevant non-immune cellular component of TME is represented by the peri-tumoral adipocytes, which are also known as cancer-associated adipocytes (CAAs). ECM proteins and various cytokines, chemokines, and hormones (i.e., adiponectin, resistin, visfatin, oestrogens, tumor necrosis factor-alfa (TNF-α), IL-1β, IL-6 and IL-8, IGF1, VEGFA, FGF2, hepatocyte growth factor, angiopoietins, CCL2 and CCL5, commonly mentioned as adipokines) may directly sustain peri-tumoral angiogenesis through the production of high levels of proteases [17,22,23]. To this regard, it has been reported that tumors that are inoculated in adipose tissue show a more elaborated vascular network than those subcutaneously injected, thus proposing a possible role of adipocytes in accelerating angiogenesis [24]. In addition, TME hosts MSCs that in physiologic conditions represent multipotent stem cells that are distinguished by the expression of multiple markers, as CD29 (named also β1 integrin), CD44, CD73, CD90, CD105, STRO, and the release of vimentin, and are able to differentiate into fibroblasts, adipocytes, pericytes, osteocytes, and chondrocytes [25]. Within the tumor environment, MSCs stimulate angiogenesis through VEGF release, and show immunoregulatory functions through immune receptors that strongly regulate the tissue microenvironment. Indeed, VEGF represents a key factor in tumor progression, when the cancer cells acquire an invasive behavior and establish a dynamic crosstalk with stromal cells stimulating strong intratumoral angiogenesis, leukocyte infiltration, fibroblast proliferation, and ECM deposition [22]. In the complex process of tumor angiogenesis, a pivotal role is also played by ECs and pericytes, the main cellular components of vessels walls. ECs represent the stromal regulators of cell proliferation, invasion, secretion of inflammatory and growth mediators, and metastatic spread through the production of TGF-β1, periostin, and VEGF, which break down ECM and lead the growth of new vascular sprouts in response to VEGFA [17]. Pericytes, which differentiate from mesenchymal precursors as ECs, are mainly recruited into tumors by PDGF-β. In tumor tissue, pericytes express high levels of α-smooth muscle actin, desmine, chondroitin sulfate proteoglycan 4 (also known as NG2), 3G5 antigen, PDGFR-β, and endosialin. These factors mainly modulate the magnitude of immune response and prevent lymphocytes extravasion and activation in tumor tissue. 

### 2.3. Development of the Pre-Metastatic Niche (PMN)

Factors that are produced by the stromal components, such as VEGFA, and molecules that are released by tumor cells and immune cell populations, including inflammatory cytokines and chemokines (e.g., TGF-β and TNF-α), represent the main players in conditioning PMNs in distant organs that are favorable to the survival and outgrowth of the recruited cancer cells [26]. Overall, PMNs are the result of the combined effects of tumor-secreted factors and tumor-shed extracellular vesicles (EVs), which promote their initiation and development. To this regard, it has been reported that cancer cell-derived EVs are an important component of PMNs transferring their cargo comprising genetic material (DNA, mRNA, and miRNA), metabolites (lipids and small molecules), and proteins. Additionally, platelet-derived EVs have been reported to induce angiogenesis and metastasis in lung and breast cancers [27,28]. A recent study has reported that in pancreatic cancer, beyond promoting tumor progression, PMN induces tumor dormancy at the liver metastatic site promoting fibrosis [3]. Once established, PMN is responsible for extravasation, colonization, and metastatic outgrowth of tumor cells [29]. Importantly, PMNs also include atypical immune cells that are recruited from bone marrow [26]. 

As a whole, the non-immune cells of TME control tumor progression and metastasis through direct and immune-mediated signals. In fact, their activity is crucial in determining the composition of immune cell infiltrate and the complex reciprocal interactions determining the immunosuppressive TME. 

## 3. The Immunosuppressive Landscape of TME

The element that mostly distinguishes TME is the multifactorial nature of immune cells. Several immune cell types with immunomodulatory activities, including TAMs, myeloid-derived suppressor cells (MDSCs), regulatory T cells (Tregs), and dendritic cells (DCs), as well as effector immune cells, such as T lymphocytes and Natural Killer (NK) cells, determine the immune landscape of TME [30]. 

### 3.1. TAMs as Major Drivers of Immunosuppressive TME

TAMs are one of the most critical immune components determining tumor fate, since the dominant population of immune cells that migrate into tumor niche are macrophages [31]. Within TME, macrophages can become TAMs, promoting tumor progression, stimulating tumor cell growth, invasion, and metastasis. The immunosuppressive role of intratumoral TAMs results from the dominance of M2-like TAMs over M1-like TAMs, which is characterized by different phenotype and function. M1 macrophages exhibit antitumor effects acting as driver of inflammatory response, thus stimulating the release of pro-inflammatory cytokines and factors, such as TNF-α, IL-6, IL-1β, IL-12, IL-15, IL-18, reactive nitrogen, and oxygen intermediates. On the contrary, M2 macrophages own potent pro-tumor activity producing a high amount of anti-inflammatory cytokines, including CCL18, IL-10, VEGF, and MMPs, thus determining the recruitment and activation of immunosuppressive cells, such as Treg and MDSCs, as well as the inhibition of T cell responses, thus promoting tumor invasion and metastasis (Figure 1A). Intratumoral TAM accumulation is sustained by resident macrophages and recruitment of circulating monocytes whose differentiation is dictated by environmental signals. Immunosuppressive cytokines, such as TGF-β, IL-4, IL-10, and IL-13, induce M2-like TAMs through monocyte polarization and macrophages differentiation. Of note, these cells are characterized by the expression of PD-L1 and they promote tumor growth through IL-10 production and PD-L1/PD-1 axis activation (Figure 1B) [32]. Nevertheless, TAMs are extremely plastic immune cells that can undergo a phenotypic switch towards M1-like TAMs upon inflammatory cytokine availability within TME [33]. As a whole, the high frequency of M2-like TAMs at the tumor site associates with poor prognosis, whereas M1-like TAM prevalence correlates with a better prognosis, although these latter can also promote malignant transformation by inducing chronic inflammation [34]. With respect to the interactions between TAMs and cancer cells, there is a bidirectional positive crosstalk: immunosuppressive cytokines that are produced by cancer cells contribute to shaping phenotype and function of M2-like TAMs, which in turn promote TME remodeling into a chronic immunosuppressive status favorable to the recruitment of other immunosuppressive immune populations, thus ultimately sustaining the growth of cancer cells. 

### 3.2. The Role of MDSCs

Another key heterogeneous immune cell population shaping TME is represented by MDSCs. These cells derive from myeloid progenitors and immature myeloid cells, whose frequency is enormously increased in inflamed conditions, as it is the intratumoral milieu [35]. Diverse cytokines and factors, including VEGF, granulocyte-macrophage colony stimulating factor (GM-CSF), IL-6, IL-10, TGF-β, interferon (IFN)-γ, IL-1β, and CCL2, are main attractors of MDSCs toward tumor tissue and affect their activation (Figure 1C) [36]. MDSCs can be divided into two main groups designated as monocytic MDSCs (M-MDSCs), which are morphologically and phenotypically similar to monocytes, and polymorphonuclear MDSCs (PMN-MDSCs), also known as granulocytic MDSCs (G-MDSCs). M-MDSCs, which have higher suppressive activity than PMN-MDSCs, play a major role in depicting TME immunosuppression. M-MDSCs are phenotypically and morphologically similar to monocytes and they accumulate in the tumor where they produce high levels of nitric oxide (NO), Arginine (Arg)-1 and IL-10, suppressing both antigen-specific and non-specific T cell responses. Within TME, M-MDSCs become more suppressive and own the peculiar trait to rapidly differentiate to TAMs [37,38]. On the contrary, PMN-MDSCs, which are primarily located in peripheral tissues, closely resemble human peripheral blood neutrophils, being CD33^+^ CD11b^+^ HLA-DR^−^ and arginine Arg-1^+^. This cellular component is mainly devoted to suppress antigen-specific T cell responses by reactive oxygen species (ROS) production [37]. Therefore, the percentage of tumor infiltrating PMN-MDSCs is very low. One of the most important mechanisms by which MDCSs inhibit T cells is the recruitment of Treg cells into tumors, as well as the stimulation of their local expansion. MDSCs, through TGF-β production, also impair B cell responses and promote the recruitment of tumor-associated neutrophils [39]. Like TAMs, MDSCs are also very plastic immune cells whose immunosuppressive activities are dynamically shaped by continuous multiple microenvironmental signals. In this light, the crosstalk between MDCSs and CSCs is crucial for supporting MDSC immunosuppressive activities and CSC maintenance strongly favoring the metastasis process. On one hand, CSCs produce factors, such as macrophage migration inhibitory factor, which are able to inhibit MDSCs apoptosis and increase the production of the immunosuppressive enzyme Arg-1 [40]. In return, metabolic enzymes, such as Arg-1, indoleamine 2,3-dioxygenase 1 (IDO1), iNOS, and NO synthase-2, as well as immunosuppressive molecules, namely NO, ROS, kinurenines, prostaglandin E (PGE)-2, IL-10, and TGF-β, which are produced by MDSCs, modulate and impair function and trafficking of effector T cells sustaining CSC tumorigenesis and drug resistance [41]. Further, MDSCs have been reported to produce CCL3, CCL4, and CCL5 chemokines favoring tumor recruitment of Treg cells [42]. Importantly, some prevalent factors within TME, such as, IFN-γ, MCSF, and VEGF, stimulate the up-modulation of PD-L1 on MDSCs, strengthening their capacity to induce the apoptosis of CD8^+^ T cells (Figure 1C). This finding strongly supports the great potential of PD-L1 blockade to release the MDSC-mediated inhibition of T cell activation [43].

### 3.3. Tregs-Mediated Immunosuppression of TME

In this immunosuppressive context, Treg cells are one of the most ready immune populations to infiltrate into TME [44]. Human Treg cells are composed by two main different subpopulations that are defined by different markers. The first is represented by FoxP3^lo^CD45RA^+^CD25^lo^ naïve/resting Treg cells that leave the thymus endowed with a weak immunosuppressive function and differentiate into FoxP3^hi^CD45RA^−^CD25^hi^ effector/activated Treg cells (eTreg cells) following T-cell receptor (TCR) stimulation. These terminal differentiated Treg cells represent the second Treg subpopulation and are owed to stronger immunosuppressive activities. Responding to chemokine gradients, Treg cells easily accumulate into TME where preferentially recognize cognate antigens undergoing activation, proliferation, and acquisition of strong immunosuppressive functions [45]. Once into tumor bed, Treg cells contribute to maintain an immunosuppressive environment through various mechanisms dampening antitumor immunity. eTreg cells fine-tune the cytokine milieu by consuming IL-2 through high CD25 expression, thus limiting the amount of this cytokine available for effector T cell proliferation and activation. Further, Treg cells produce cytokines with immunosuppressive activity, such as IL-10, TGF-β and IL-35, which directly inhibit T cell function and cytotoxic factors, namely perforin and granzyme, which are able to kill effector T cell [46]. Within TME, chemokines that are produced by Treg cells, along with other TME factors, such as IL-6, M-CSF, IL-13, IL-1β, and VEGF, also determine DC dysfunction through several mechanisms [47]. While IL-6 switches the differentiation of monocytes from DC to macrophages and sustain DC tolerogenic phenotype [48], IL-10 suppress DC functions converting immunogenic into tolerogenic DCs resulting in anergic cytotoxic CD8^+^ T cells induction [49], and VEGF, TNF-β, IL-13, and IL-1β inhibit DC differentiation switching progenitors towards MDSCs and TAMs [50].

Importantly, Treg cells express a series of immune checkpoint molecules, including cytotoxic T-Lymphocyte Antigen 4 (CTLA-4), TIM-3, PD-1, and glucocorticoid-induced TNFR-related protein (GITR), through which they suppress the functions of DCs and effector T cells. For example, the engagement of CD80/CD86 on DCs by CTLA-4 on Treg cells stimulate the conversion into tolerogenic DCs highly producing IDO, which in turn dampen effector T cells and promote Treg function (Figure 1D) [51]. 

### 3.4. The Immunosuppressive Plasticity of DCs

Within TME, tumor-infiltrated DCs often promote immunosuppression and tolerance instead of driving antitumor immunity. A heterogeneous population comprising two major subsets with different functions represents DCs: conventional DCs (cDCs) and plasmacytoid DCs (pDCs). In turn, cDCs consist of two subtypes, BDCA3^+^ cDC1s, specialized in cross-priming CD8^+^ T cells, and BDCA1^+^ cDC2s, efficient in presenting antigens on MHC-II to CD4^+^ T cells. Many chemokines, including CXCL1, CCL2, CCL20, CCL4, CCL5, and XCL1, which are produced by both cancer and immune cells, drive BDCA3^+^ cDC1 recruitment into TME with the potential to induce local cytotoxic T cell function. However, the immunosuppressive milieu of TME reeducates BDCA3^+^ cDC1 in losing their strong capability of antigen cross-presentation to CD8^+^ T cells. For example, IL-10 suppresses IL-12 production and TGF-β inhibits antigen-presenting functions of BDCA3^+^ cDC1 [52]. Moreover, the expression of inhibitory molecules, such as PD-1 and TIM-3, suppress the capability of cDCs to stimulate CD8^+^ effector T cells [53,54]. Importantly, immunosuppressive TME mediators, such as PGE-2 and TGF-β, alter also the functional activities of BDCA2^+^BDCA4^+^CD123^+^ pDCs, which results in the acquisition of the capability to induce T cell-suppressive activity [55]. Overall, the plasticity of DCs within TME underlies how tumors may play with the Janus face of immune cells to take advantage for tumor progression and metastasis spread. 

### 3.5. The Role of Neutrophils

The complex array of interactions undergoing into tumor sites is also crucial for the function of neutrophils, another heterogeneity immune cell population whose regulatory function in cancer progression and metastasis has been largely neglected [56]. These cells may differ in phenotype and function, and comprise the N2-subtype with pro-tumoral functions and the N1-subtype, owing immuno-activating properties. Into tumor tissues, neutrophils exhibit similar morphology and surface marker expression of PMN-MDSCs, but they own less suppression capacity towards T-lymphocytes. Several regulators, including G-CSF, TGF-β, and IL-17, produced by either cancer or immune cells, stimulate neutrophils recruitment into tumor where they prevalently polarize towards the N2-subtype, contributing to educate immunosuppressive immune system. For instance, neutrophils produce TGF-β and iNOS that promote M2-like TAM transition and CD8^+^ T cell suppression [57].

### 3.6. CD8^+^ T Cells as the Main Players of Antitumor Response

Within the complex immunosuppressive contexture of TME, CD8^+^ T cells are supposed to act as the main execute actors of antitumor response. However, tumor-specific CD8^+^ T cells need to successfully trafficking into tumor, proliferate, differentiate, and acquire competent functions in order to induce a competent immune response. Differentiation and tumor antigen encounter determines the production of effector CD8^+^ T cells, including memory and cytotoxic T cells. While central memory T cells locate into lymph nodes, where they are reactivated upon secondary exposure to antigens, effector memory T cells own cytotoxic properties and circulate through tissues, and resident memory T cells persist in pathogenic tissues [58]. Several factors, including chemokines, cytokines, and co-stimulatory and inhibitory molecules, drive T cell homing to tumor tissues. As an example, both the down-regulation of CD62L on CD8^+^ T cells and IL-12 reduction promoted by MDSCs impair T cell trafficking into tumor [59]. Likewise, M2-like TAMs produce many factors, including VEGF, which negatively regulate the expression of the adhesion molecules ICAM-1 and VCAM-1 reducing CD8^+^ T cell trafficking from the circulatory system to the tumor site [60]. Once into TME, the activation of CD8^+^ T cells occurs through a three-step process that include antigen presentation by DCs, or other antigen presenting cells (APCs) in the phase of re-stimulation, the delivery of a costimulatory signal from DCs, and the stimulation from extracellular cytokines. Generally, the immunosuppressive TME is unable to ensure all of these conditions and favors CD8^+^ effector T cell dysfunction (Figure 1D). As discussed above, TME employs an array of strategies to modify many immune cells towards immunosuppressive phenotype and function, thus leading to insufficient T cell stimulation. In this game, immunosuppressive factors that were produced by MDSCs, as well as by Treg, block the function of DCs, which in turn become endowed with immunosuppressive function, such as production of large amount of IDO, and unable to stimulate CD8^+^ T cells. In addition, the activation of immunosuppressive pathways, such as the conversion of tryptophan into kynurenine, limits T cell proliferation and activation, whereas stimulate Treg production. Nevertheless, inhibitory signals plays a pivotal role in determining T cell dysfunction, as many immune cells, such as M2-like TAMs, expresses high levels of surface molecules, including PD-L1, PD-L2, CD80, and CD86, which restrict CD8^+^ T cell activities upon binding to the immune-checkpoint receptors, PD-1 and CTLA-4. Lastly, other molecules, such as the ectonucleotidases CD39 and CD73 that mediate the conversion of extracellular adenosine triphosphate or adenosine diphosphate to adenosine monophosphate and then the production of the immunosuppressive adenosine, strongly inhibit CD8^+^ T cell function [61]. As a result, the complex immunosuppressive network of TME is fully armed to prevent the function of CD8^+^ T cells disabling their ability in destroying tumor cells.

## 4. The Strength of Targeting TME with Immunotherapy

Given the potential to re-educate the nature of the dynamic TME towards an antitumor immunity, in the past few years many efforts have been dedicated to design new efficacious immunotherapeutic strategies that are able to target non-immune and immune components of TME. 

### 4.1. Immune-Checkpoint Inhibitors Therapies

The administration of immunotherapies that target the immune-checkpoint receptors CTLA-4, PD-1, and PD-L1, which are widely defined as immune-checkpoint inhibitors (ICIs), have shown a significant improvement of the overall survival (OS) of patients that are affected by metastatic melanoma, Hodgkin lymphoma, non-small-cell lung cancer (NSCLC), head and neck squamous carcinoma, Merkel cell carcinoma, and many others types of cancer [62]. The use of CTLA-4-blocking antibodies, named ipilimumab and tremelimumab, is able to prevent the competition of this inhibitory molecule with CD28 for binding to CD80 and CD86, which releases the proliferation and activation of T cells following the recognition of specific tumor antigen, mainly in lymph nodes (Figure 2A) [63,64]. However, even if the survival of patients with metastatic melanoma treated with ipilimumab was improved, and consistently this therapy received the approval from FDA in 2011, toxicities and inflammatory adverse events were still observed [65,66]. 

Following the important therapeutic advance with CTLA-4 blocking, a breakthrough in immunotherapy has been achieved with the development of inhibitors of the PD-1 receptor and of its ligand, PD-L1. While the expression of PD-1 is promoted by TCR triggering and it induces the release of pro-inflammatory cytokines by activated T cells, the persistent activation of the PD-1/PD-L1 axis determines T cell exhaustion [67]. Therefore, by blocking this event, anti-PD-1 and anti-PD-L1 antibodies promotes T cell proliferation, which restores the antitumor cytotoxic T cell response (Figure 2A). Of interest, it has been found that the clinical outcome of ICI treatments is associated to several intrinsic TME biomarkers: i) the variable expression of PD-L1 by tumor cells or infiltrating immune cells; ii) high tumor mutational level; and, iii) enhanced densities of tumor-infiltrating lymphocytes (TIL). Consistent with the high expression of PD-L1, clinical studies that were carried on patients with melanoma, NSCLC, urothelial cancer, renal carcinoma, head and neck squamous cell carcinoma and gastric cancer treated with these check-point blockade antibodies showed an overall response rate (ORR) that ranged from 30% to 53% [3,68]. Other studies demonstrated that there is a direct correlation between the highest response rates (RR) to anti-PD-1/PDL-1 therapies and high mutational loads identified in melanoma, NSCLC, gastric cancer, and bladder cancer [69,70]. Accordingly, several reports indicated that the treatment with anti-PD-1 was strongly efficacious in patients with tumors highly expressing PD-L1 and with high mutational levels [71,72,73]. Likewise, high density of TIL (CD3^+^, CD8^+^, and CD45RO^+^ memory) within TME was associated with a good response in patients with colon cancer and melanoma treated with anti-PD-1 or PD-L1 therapies [74,75,76].

Given that in many types of tumors ICIs have limited efficacy, although much higher than conventional treatments, their antitumor efficacy has been exploited by combination treatments. The combination of ipilimumab with nivolumab has led to a significant improvement of the RR and survival rates (SR) in patients with metastatic melanoma, colorectal carcinoma characterized by DNA mismatch repair or microsatellite instability, metastatic renal cell carcinoma (mRCC), and advanced NSCLC [77]. However, this therapeutic combination was associated with severe adverse events, whose toxicity need to be reduced by further studies. Given that the angiogenic features tightly associate to the immune competence of TME, new therapeutic regimens have been recently designed to evaluate the synergistic antitumor effects of anti-VEGF monoclonal antibodies (bevacizumab) in combination with CTLA-4 or PD-L1 blockers. Data from metastatic melanoma patients treated bevacizumab, and ipilimumab showed that the significant improvement of the OS as associated with the capability of this drug combination to induce upregulation of CD31, E-selectin, VCAM-1, and other adhesion molecules significantly increasing the trafficking of mature DCs and cytotoxic T cells into TME [78,79]. Bevacizumab or anti-angiogenesis TKI (axitinib) have also been associated to several anti-PD-L1 antibodies (atezolizumab, avelumab, nivolumab, pembrolizumab) to treat several types of solid tumors [80]. Of interest, the anti-angiogenic treatments as monotherapy were found to significantly increase the ratio of anti/pro-tumor immune cells, while reducing the expression of immune checkpoints molecules within TME, whereas their combination with ICIs also reactivated a competent immune-response and normalized TME vessel structure associated with improved patient OS [29,81]. However, also in these cases, further studies are needed to reduce toxic adverse effects of combination therapy. 

### 4.2. Cancer Vaccines in Combination with ICIs and Other Antitumor Therapies

Attempts to target the complex immunosuppressive niche of TME have been made by the use of various cancer vaccines. 

Since one of the major trait of TME is the dysregulation of DCs, often being unable to undergo correct differentiation, activation, and antigen presentation, DC-based vaccines represent a promising therapeutic approach to stimulate antitumor immunity [82]. Therefore, also due to their low risk of toxicity, DC-based cancer vaccines have been widely explored in the past two decades. The first therapeutic DC cancer vaccines were based on the use of monocytes (mo-DC) obtained from the peripheral blood cultured and differentiated with GM-CSF and IL-4. Following pulsation with tumor associated antigens (TAA), such as peptides, proteins, DNA, virus, mRNA, lysate or tumor cells, the immature DCs were maturated with other growth factors, cytokines, or toll-like receptor agonist (lypopolysaccaride), and then re-infused in the patient (Figure 2B) [83,84]. Further pre-clinical and clinical studies using allogeneic mo-DC obtained from healthy histocompatible donors or dendritic cell-tumor cell hybrids have been performed to improve the efficacy of vaccination using autologous mo-DC [85]. Further implementation of the DC-based vaccine design, including also the use of mo-DC differentiated with IFN-α, have demonstrated that these advanced therapy medicinal products may represent promising antitumor strategy [86,87]. 

Given the strong component of tumor angiogenesis in the immune escape at TME level, the combination of DC vaccines with anti-angiogenic therapies has also been exploited. Although a phase II clinical study combining a personalized DC-based vaccine and the VEGF tyrosine kinase inhibitor sunitinib induced immunological responses and prolonged the survival of patients with mRCC, the subsequent phase III trial was not successful. In spite of that, other clinical studies that explored the efficacy of cancer vaccines and antiangiogenic therapies are currently running [80].

Nowadays, many on-going studies are challenging the combination of DC-vaccines with ICIs with the goal of targeting complementary immune signals, thus to enhance the number and improve the activity of tumor-specific T cells infiltrating TME [88,89]. Preliminary clinical data indicate that the combination of ipilimumab with GVAX vaccine platform (GVAX-Pancreas NCT00084383) has enhanced OS in patients with pancreatic adenocarcinoma as compared to ipilimumab alone [90]. Furthermore, a phase II study on advanced melanoma patients treated with TriMixDC-MEL, autologous monocyte-derived DCs electroporated with synthetic mRNA, in combination with ipilimumab, demonstrated an efficacious tumor response, which is characterized by eight complete responses and seven partial responses [91]. Moreover, a vaccine that is based on overlapping HPV E6 and E7 peptides in combination with nivolumab administered to 24 patients with incurable HPV-16–positive cancer showed an ORR of 33% and a median OS of 17.5 month as compared to anti-PD-1 alone [92].

### 4.3. CAR T cells Therapies

The adoptive cell therapy that is based on the use of chimeric antigen receptor (CAR) T cells represents the most novel and promising personalized treatment for cancer patients and many efforts are currently dedicated to its improvement. The first generation of CAR T cells was developed through the engineering of patient T cells to express antigen-binding fragments of a specific antibody that was fused to a transmembrane domain and a CD3ζ chain for T cell activation (Figure 2C). In the second generation one or two intracellular co-stimulatory signalling domains were added (CD28 and CD137 or ICOS and 4-1BB) [93]. Of note, the reinfusion of CAR T cells targeting CD19 led to a RR of 80% in adult patients with relapsed/refractory B-cell acute lymphoblastic leukemia and lymphoma, which are usually characterized by a very bad/poor prognosis [94,95,96]. Unfortunately, the use of these therapies did not reach similarly good results for the cure of solid tumors, mainly due to: (i) the complexity in selecting specific TAAs to efficiently target cancer cells while avoiding cross-reactive toxicity, (ii) difficulties of CAR T cells to penetrate into the TME thus overcoming its immunosuppressive intrinsic features, and, (iii) the maintenance of efficacy and persistence. The identification of unique target antigens is absolutely required in order to efficiently target cancer cells within TME. However, this process is quite complex, because the wide and low expression of many tumor antigens. In fact, the infusion of CAR T cells targeting human epidermal growth factor receptor 2 (HER2) in metastatic colon cancer patients resulted in fatal events for some patients, because low levels of HER2 are also expressed on the lung epithelial cells that were hit by the CAR T cells [97]. Similarly, fatal encephalitis occurred in neuroblastoma patients receiving anti-GD2 CART T cells due to the expression of GD2 in the brain [98]. To avoid the persistence of CAR T cells and to control their potential reactivity towards normal cells, a third generation of CAR T, including suicide genes, such as inducible caspase 9 or truncated EGFR inducing antibody-mediated cell death, has been generated [99]. 

Taking advantage of immunoproteomic technologies, advances have also been made in the discovery of new proteins (neoantigens) or peptides (neoepitopes) that can be useful TAAs for designing CAR T cells against a variety of tumors, such as colon cancer, melanoma, and glioblastoma multiforme [100]. Furthermore, CAR T cells have been engineered to recognize two antigens instead of one (dual or tandem CARs), so to better cover the heterogeneous distribution of cancer antigens. To this regard, CAR T cells expressing both HER2 and mucin 1 showed promising results in breast cancer preclinical studies, while CAR T cells that targeted HER2 and IL-13Rα2 were more efficacious than CAR targeting only one antigen in a pre-clinical glioma model CD19/CD20. Of note, clinical trials with dual CAR targeting CD19/CD20 and CD19/CD22 are currently running [100].

CAR T cells have also been generated to induce epitope spreading leading to the release by cancer cells of specific tumor neoantigens or epitopes that are then presented by APC to TILs, thus triggering a secondary immune TILs antitumor response. To this extent, an antitumor immune response suggesting epitope spreading was observed following mesothelin-specific mRNA CAR T cells treatment in a case study on two patients with malignant pleural mesotheliomas and metastatic pancreatic cancer, respectively [101].

Several experimental combinational treatments were designed to overcome the low efficacy of CAR T cells to infiltrate TME. While considering that the immature and not regular feature of the tumor vessels could obstacle the infiltration of T cells into the tumor tissue, the combination of CAR T cell therapy with anti-angiogenic therapy targeting VEGF or other endothelial receptors, such as CD276 or endothelin B, can represent a good option [100]. Moreover, given the high levels of chemokines that are present in TME, clinical studies based on the use of CD2/CCR2b engineered CAR T cells and mesothelin CAR/CCR2 T cells showed a migration and homing increase of the CAR T ranging bethween 10- and 12-fold into neuroblastoma tumors and malignant pleural mesothelioma, respectively [102,103].

However, from the TME point of view, the most efficacious combination therapies using CAR T cells are those with ICIs that have demonstrated a significant improvement of CAR T cell persistence into the tumor due to the reversion of immunosuppressive environmental signals [104].

## 5. Conclusions

In the era of personalized medicine, the deep knowledge of TME immunosuppressive signals causing tumor progression and metastasis need to be better defined to design novel and more efficacious therapeutic strategies that also overcome the resistance that often occurs. 

Clearly, the dynamic crosstalk among the multiple cellular components of TME mediated by a complex net of cell-cell interactions and a heterogeneous array of soluble factors depicts novel pathways that are to be targeted. Dissecting the reciprocal interactions among the large variety of tumor factors, immune and stromal cells will allow for fully understanding the immunosuppressive TME, which govern tumor growth and metastasis escape. Although it is extremely complex to explore the crosstalk among different cell populations, studying the immunosuppressive mechanisms that were adopted by the single components of TME limits the comprehension of those features that may change in different conditions, such as primary and metastatic sites or different types of tumors. Only the discovery of intra and inter-tumor heterogeneity in the dynamic crosstalk of its components can allow for identifying reliable biomarkers that are useful for stratify the patients to be treated with personalized therapeutic strategies. To this regard, the remarkable advances in methods and techniques will be of great help for patients’ stratification not only based on the tumor type, but also on the specific characteristics of the TME. Importantly, this approach will also allow also for elucidating the continuous changes that occur within TME during therapy, providing information that is useful for identifying new therapeutic targets. Most importantly, the immune-modulating therapeutic combinations, which have recently showed promising results, can be further optimized to better block TME immune suppressive pathways and stimulate antitumor immunity. In this light, the assessment of the dynamic niche of TME remains an important challenge, which is fundamental for improving the outcome of cancer patients and for the implementation of the strength and cost-effectiveness of antitumor strategies.

## Figures and Tables

**Figure 1 ijms-20-04320-f001:**
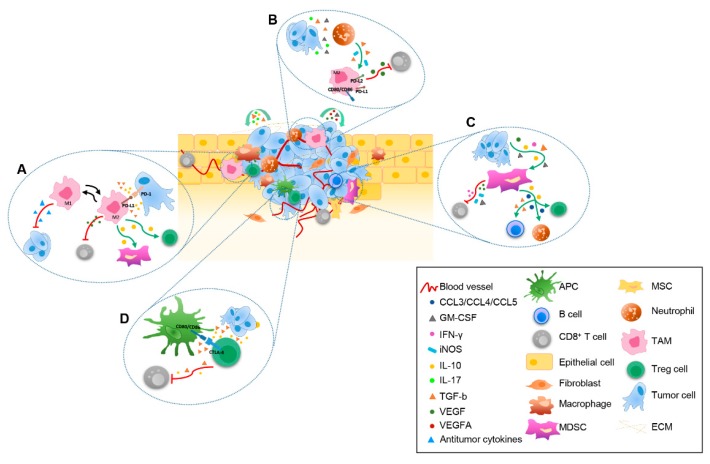
The dynamic crosstalk within tumor microenvironment (TME). Schematic representation of the main mechanisms underlying the interaction among extracellular matrix (ECM), stromal cells, tumor cells and infiltrating immune cells driven by released immunosuppressive cytokines and chemokines. The following dynamic interactions between cellular components are indicated: (**A**) antigen presenting cells (APC), tumor cells, regulatory T cell (Treg) and CD8^+^ T cells; (**B**) tumor cells, neutrophils, tumor-associated macrophages (TAM) and CD8^+^ T cells; (**C**) tumor cells, TAM, Treg cells, myeloid-derived suppressor cell (MDSC) and CD8^+^ T cells; (**D**) tumor cells, MDSC, CD8^+^ T cells, B cells, neutrophils and Treg cells.APC, Antigen presenting cell; ECM, extracellular matrix; MDSC, myeloid-derived suppressor cell; MSC, mesenchymal stem cell; TAM, tumor-associated macrophage; Treg, regulatory T cell.

**Figure 2 ijms-20-04320-f002:**
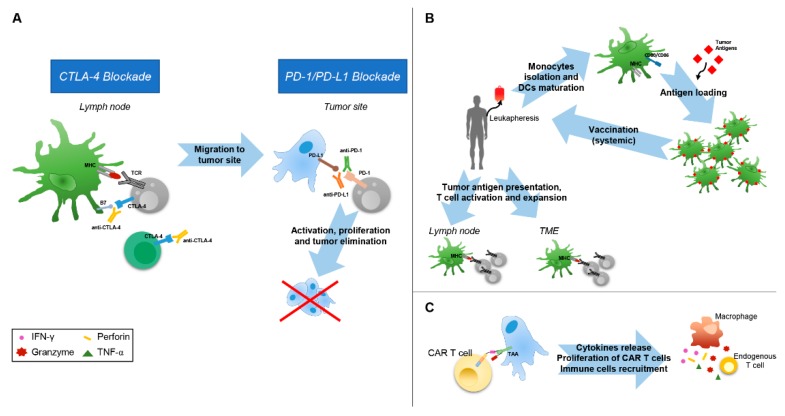
Mechanisms of action of immunomodulating agents. (**A**) CTLA-4 and PD-1 pathway blockade. CTLA-4 blockade allows activation and proliferation of tumor-specific T cells and reduces Treg-mediated immunosuppression. PD-1 pathway blockade reestablishes the activity of quiescent tumor-specific T cells and stimulates T cell migration and proliferation. (**B**) To generate DC cancer vaccines, DCs are harvested from patients by leukapheresis, maturated ex vivo and, in some cases, loaded with tumor antigens before reinfusion into patients. At TME and lymph node levels, DCs present tumor antigens to tumor-specific T cells, resulting in T cell activation and expansion. (**C**) CAR T cells carry modularly constructed fusion receptors that recognize specific antigen on cancer cells; upon binding, the intracellular signaling domains induce signal transduction cascades stimulating antitumor T cell activities. CTLA-4, cytotoxic T-lymphocyte–associated antigen 4; DC, dendritic cell; MHC, major histocompatibility complex; PD-1, programmed death 1; PD-L1, programmed death ligand 1; TCR, T-cell receptor; TAA, Tumor associated Antigen.

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
