# Peer review of "The Janus Face of Tumor Microenvironment Targeted by Immunotherapy"

_ijms, 2019, doi:10.3390/ijms20174320_

Round 1

Reviewer 1 Report

The work is extremely important to ongoing trials on immunotherapy of cancer. The review is very well written with clarity. Tumor microenvironment and its interaction of with immune systems are documented are well documented. 

Author Response

Response to Reviewer 1 Comments

Point 1: The work is extremely important to ongoing trials on immunotherapy of cancer. The review is very well written with clarity. Tumor microenvironment and its interaction of with immune systems are documented are well documented.

Response 1: We really appreciate the positive comments of the reviewer and we wish to thank her/him for having highlighted the relevance of the topics discussed in our review. According to the suggestion, we have carefully revised the manuscript for English mistakes or inaccuracy.

List of corrections/modifications

Lane51: “between” has been eliminated

Lanes 159-162: the sentence has been modified has follow “Factors produced by the stromal components, such as VEGFA, and molecules released by tumor cells and immune cell populations, including inflammatory cytokines and chemokines (e.g. TGF-β and TNF-α), represent the main players in conditioning PMNs in distant organs that are favorable to the survival and outgrowth of recruited cancer cells”

Lanes 301-303: the sentence has been changed in “Overall, the plasticity of DCs within TME underlies how tumors may play with the Janus face of immune cells to take advantage for tumor progression and metastasis spread”

Lane 321: “execute players” has been modified in “execute actors”

Lane 537: “to better blockade” has been modified in “to better block”.

Reviewer 2 Report

The present review article discusses the recent knowledge on the immunosuppressive crosstalk within the tumor microenvironment and puts forward new perspectives on how immunotherapeutic approaches may exploit tumor immune signals to generate antitumor immunity.

It is an interesting and important topic, the manuscript is well-written with a lot of up-to-date information, however, some points should be addressed:

a. Please improve the abstract, explaining the Janus face

Line11: thought?? cell cell interactions .... please correct

b.    Part 2. (lines 46-152) and Part 3. (lines 153-303) are very long and not easy to follow. Please split it in parts using subtitles for better understanding.

c. Line 472: 'extraordinary advance'.... please change to 'remarkable advances..'

d. Line 477: 'blockade..' please change to 'block' or 'inhibit'

Author Response

Response to Reviewer 2 Comments

The present review article discusses the recent knowledge on the immunosuppressive crosstalk within the tumor microenvironment and puts forward new perspectives on how immunotherapeutic approaches may exploit tumor immune signals to generate antitumor immunity.

It is an interesting and important topic, the manuscript is well-written with a lot of up-to-date information, however, some points should be addressed:

Point 1: Please improve the abstract, explaining the Janus face.

Response 1: We thank the reviewer for the precious advice. Accordingly, we have improved the abstract deepening the mentioned aspect.

Point 2: Line11: thought?? cell cell interactions .... please correct.

Response 2: We apologize for this inaccuracy that has been corrected.

Point 3: Part 2 (lines 46-152) and Part 3 (lines 153-303) are very long and not easy to follow. Please split it in parts using subtitles for better understanding.

Response 3: We thank the reviewer for this appropriate criticism. As suggested, we have divided the two mentioned sections (Part 2 and Part 3) in subparagraphs with subtitles for better understanding.

Point 4: Line 472: 'extraordinary advance'.... please change to 'remarkable advances..'

Line 477: 'blockade..' please change to 'block' or 'inhibit'

Response 4: We thank the reviewer for these observations. Accordingly, we have modified the text as suggested.

We have also carefully revised the whole manuscript for English mistakes or inaccuracy.

List of corrections/modifications

Lanes 10-25: Abstract has been corrected as follow “The tumor microenvironment (TME) is a complex entity where host immune and non-immune cells establish a dynamic crosstalk with cancer cells. Through cell-cell interactions, mediated by key signals such as the PD-1/PD-L1 axis, as well as the release of soluble mediators, this articulated process defines the nature of TME determining tumor development, prognosis and response to therapy. Specifically, tumors are characterized by cellular plasticity that allows microenvironment to polarize towards inflammation or immunosuppression. Thus, the dynamic crosstalk among cancer, stromal and immune components crucially favors the dominance of one of the Janus-faced contexture of TME crucial to the outcome of tumor development and therapeutic response. Mostly, however, TME is dominated by an immunosuppressive landscape that blocks antitumor immunity and sustain tumor progression. Hence, in most cases the immunosuppressive components of TME are highly competent in suppressing tumor-specific CD8+ T lymphocytes, the effectors of cancer destruction. In this complex context, immunotherapy aims to arm the hidden Janus face of TME disclosing and potentiating antitumor immune signals. Herein, we discuss the recent knowledge on the immunosuppressive crosstalk within TME, and share perspectives on how immunotherapeutic approaches may exploit tumor immune signals to generate antitumor immunity.”

Lane 11: “Through cell-cell interactions” has been corrected

Lane51: “between” has been eliminated

Lanes 60-64: the section has been modified as follow

“This heterogenous microenvironment is known as TME (Figure 1).

2.1 The role of ECM

The complex interactions between tumor cellular components and ECM may directly...”

Lanes 95-99: the section has been modified as follow

“…extravasation and formation of the metastatic niche [13].

2.2 The contribution of stromal cells

In complex landscape of TME, tumor growth...”

Lanes 155-164: this part has been adapted as follow

“...lymphocytes extravasion and activation in tumor tissue.

2.3 Development of the pre-metastatic niche (PMN)

Factors produced by the stromal components, such as VEGFA, and molecules released by tumor cells and immune cell populations, including inflammatory cytokines and chemokines (e.g. TGF-β and TNF-α), represent the main players in conditioning PMNs in distant organs that are favorable to the survival and outgrowth of recruited cancer cells [27]. Overall, PMNs are the result of the combined effects of tumor-secreted factors and tumor-shed extracellular vesicles (EVs) that promote their initiation and development. To this regard, …”

Lanes 186-190: this section has been modified as follow:

“…determine the immune landscape of TME [31].

3.1 TAMs as major drivers of immunosuppressive TME

TAMs are one of the most critical immune components determining tumor fate, since…”

Lanes 215-219: this part has been modified as follow:

“..sustaining the growth of cancer cells.

3.2 The role of MDSCs

Another key heterogeneous immune cell population shaping TME is represented by MDSCs.”

Lanes 254-258: this section has been modified as follow:

“…inhibition of T cell activation [44].                                                                                                 3.3 Tregs-mediated immunosuppression of TME.

In this immunosuppressive context, Treg cells…”

Lanes 283-288: this section has been changed as follow:

“…promote Treg function (Figure 1D) [52].

3.4 The immunosuppressive plasticity of DCs

Within TME, tumor-infiltrated DCs often promote immunosuppression…”

Lanes 301-307: this part has been changed in

“…T cell-suppressive activity [56]. Overall, the plasticity of DCs within TME underlies how tumors may play with the Janus face of immune cells to take advantage for tumor progression and metastasis spread.

3.5 The role of neutrophils

The complex array of interactions…”

Lanes 316-320: the paragraph has been divided as follow

“…transition and CD8+ T cell suppression [58].

3.6 CD8+ T cells as the main players of antitumor response

Within the complex immunosuppressive contexture of TME, CD8+ T cells are supposed…”

Lane 321: “execute players” has been modified in “execute actors”

Lane 532: “the extraordinary advance” has been changed in “the remarkable advances”

Lane 537: “to better blockade” has been modified in “to better block”.